# Study of In Situ Foamed Fly Ash Geopolymer

**DOI:** 10.3390/ma13184059

**Published:** 2020-09-12

**Authors:** Zijian Su, Wei Hou, Zengqing Sun, Wei Lv

**Affiliations:** 1School of Minerals Processing and Bioengineering, Central South University, Changsha 410083, China; suzijian@csu.edu.cn (Z.S.); houwei@csu.edu.cn (W.H.); 2Institute of Building Materials Research (IBAC), Rheinisch-Westfälische Technische Hochschule Aachen University, Schinkelstr. 3, 52062 Aachen, Germany; 3School of Metallurgy and Environment, Central South University, Changsha 410083, China; lvweicsu@163.com

**Keywords:** fly ash, foam concrete, geopolymer, aluminum powder

## Abstract

Foamed fly ash geopolymer was synthesized in this work to produce geopolymeric lightweight concrete (GLWC). Fly ash was activated by sodium silicate solution, and aluminum powder was employed as an in situ chemical foaming agent. The synthesized pastes were cured at 40 °C for 28 days, with bulk densities of resultant GLWCs ranging from 600 to 1600 kg/m^3^. The resulting mechanical properties, thermal conductivity, microstructure, and reaction product were fully characterized. Results show that GLWC had higher mechanical strength than commercial aerated concrete and developed 80–90% of its corresponding 28 days strength after curing for 7 days. For densities from 1200 to 600 kg/m^3^, the thermal conductivity diminished from 0.70 to 0.22 W/mK, which is much better than that of its counterpart, ordinary Portland cement (OPC). Scanning electron microscopy (SEM) images revealed decent matrices comprising geopolymeric gel and unreacted fly ash.

## 1. Introduction

Lightweight concretes (LWCs) are cementitious compositions, and their bulk density (300–2000 kg/m^3^) is significantly lower than that of ordinary concretes (2200–2600 kg/m^3^) [1]. Lightweight concretes have low density, low thermal conductivity, superior acoustic insulation, and fire resistance, and they are widely applied as building materials. There are three main ways to reduce the density of LWC products: (i) removing the fine sand from the concrete mixture, resulting in voids between pieces of coarse aggregate; (ii) replacing part of the solids by air voids introduced by foaming agents (this LWC is known as aerated or foam concrete); and (iii) replacing part of the solids with a lightweight aggregate such as perlite, pumice, zeolite, or polyurethane foam [2,3,4,5,6].

In the past decade, the preparation of LWC by incorporating pozzolanic siliceous materials, especially industrial wastes such as fly ash, has attracted increasing attention because of its ecological and economical advantages [7]. Fly ash is a well-known industrial waste produced by coal-fired power plants. The total worldwide production is estimated to exceed 550 Mt per year [8]. Only a small proportion of fly ash is used in recycled forms for various purposes. By replacing part of the cement, fly ash can be used for LWC production. Kearsley et al. [9] studied the effects of adding high contents of fly ash on the performance of foam concrete, and the results showed that fly ash could replace up to 67% of cement without significant reductions in compressive strength. In addition, fly ash can be transformed to lightweight aggregate and subsequently used to produce LWC. Kockal et al. [10,11] prepared lightweight fly ash aggregate and discussed its influence on the behavior of concrete mixtures. The final LWC obtained had a density of around 2000 kg/m^3^ and exhibited compressive strength ranging from 42.3 to 55.8 MPa, with low chloride permeability and eligible durability factors.

However, the main binder of LWC in the existing literature is still ordinary Portland cement (OPC), which has a highly polluting and energy consuming production process [12,13]. The production of 1 kg of Portland cement emits about 0.8 kg CO_2_ [14], which has been considered as one of the main causes of the greenhouse effect [15]. Besides, the production of cement requires a high burning temperature (1300–1450 °C) and consumes massive amounts of energy [16]. Therefore, the development and commercialization of a sustainable binder is of high urgency to satisfy requirements of construction materials and reduce the corresponding environmental impacts. The emergence of geopolymers may be an attractive way to solve this issue.

Geopolymers are a class of inorganic materials synthesized from aluminosilicate materials and an alkaline solution. This concept was first introduced by Joseph Davidovits in the interval between the 1970s and the 1980s [17]. According to Davidovits, geopolymers are three-dimensional materials containing various amorphous to semicrystalline phases composed of oxygen–silicon–aluminum. The network of geopolymers comprises SiO_4_ and AlO_4_ tetrahedrons, which are alternately connected together in three directions by oxygen bridges [18]. In this configuration, the negative charge introduced by AlO_4_ tetrahedrons is generally balanced by alkali metals (typically Na^+^, K^+^, or both) [19]. 

Benefiting from their unique configuration, geopolymers possess advantages when compared to OPC, including better mechanical properties and durability; lower permeability; better resistance to heat, acid, and freeze–thaw; and lower heavy metal leaching content [20]. Furthermore, the synthesis of geopolymers does not need high-temperature treatment, and the CO_2_ emission is much less than that of OPC [21,22]. Such properties make geopolymers robust candidates for use as sustainable construction materials. Geopolymers can also be used as packaging materials and refractory materials.

In recent years, researchers have studied the preparation of LWC based on geopolymeric binders. Vaou et al. [23] investigated foamy geopolymers using perlite as raw material. The results showed that the final products had superior fire-resistant properties, but the strength was relatively low. Liu et al. [24] reported a method for synthesizing porous geopolymer at 80 °C, using α-Al_2_O_3_, Al powder, metakaolin, and phosphoric acid as raw materials. The compressive strength of the products was 6–13.7 MPa, and they showed excellent thermal stability. Aguilar et al. [25] found that it is possible to produce LWC based on metakaolin-based geopolymers using Al powder as a foaming agent. As an amorphous aluminosilicate material, fly ash has potential to be successfully utilized as a low-cost raw material in GLWC preparation. 

The objective of this work was to synthesize GLWC using fly ash as the raw material via geopolymer technology and employ Al powder as an in situ foaming agent. Experiments on a laboratory scale were carried out, and a series of samples with different bulk density were prepared. The mechanical, physical, and microstructural properties of the raw materials and final products were examined. The comparison between fly-ash-based GLWC and commercial aerated concrete manufactured with OPC was also conducted.

## 2. Materials and Methods 

### 2.1. Materials

The fly ash used in this study was Class F according to ASTM C618-94a [26]; it was obtained from Sinopec Jinling Petrochemical Power Plant (Nanjing, China). The chemical composition of the fly ash was determined by a 9800XP^+^ X-ray fluorescence (XRF) (Thermo Electron Corporation, Ecublens, Switzerland), and the result is shown in Table 1. Particle size distribution of raw fly ash (FA) was measured on a Mastersizer 2000 laser analyzer (Malvern, UK), showing an average particle size (d50) of 12.520 μm, as shown in Figure 1.

A sodium silicate solution of modulus 1.5 (SiO_2_/Na_2_O molar ratio of 1.5) was used as an alkaline solution for the synthesis of GLWC. The solution was prepared by dissolving an industrial grade solid sodium silicate (SiO_2_, 60.3 wt.%; Na_2_O, 26.0 wt.%; molar ratio of SiO_2_/Na_2_O, 2.38) and sodium hydroxides (analytical reagents) in water.

Al powder was used as a foaming agent for GLWC (AR agent with 99.8 wt.% purity). Al powder is reactive in an alkaline environment and will generate H_2_ gas according to the following reaction:$$2\mathrm{Al}\;+\;2\mathrm{NaOH}\;+\;2H_{2}O\;\to \;{\mathrm{NaAlO}}_{2}\;+\;3H_{2}$$

### 2.2. Sample Preparation

The samples were manufactured according to the following procedure: The alkaline solution was prepared in advance, and then the temperature of the solution naturally reduced to room temperature. The fly ash was mechanically mixed with alkaline solution for 5 min in a planetary mixer to obtain a consistent slurry. For the slurry, a suitable liquid/solid ratio of around 0.5 was employed. To complete the mixture preparation, Al powder was added to the slurry, and the mixture was mixed for another 2 min. The quantity of Al powder added into the slurry was determined by target specimen densities. The slurries were cast in molds and subsequently sealed with polyethylene film. They were then set in a laboratory oven at 40 °C and 100% relative humidity under ambient pressure for 24 h before de-molding. Then, the specimens were individually stored in sealed vessels and returned to the isothermal oven, where they were kept at 40 °C for the prescribed period of time. The obtained specimens were transferred in an electric blast drying oven and dried at 40 °C for at least 24 h before any physical and microstructural tests were conducted. In order to obtain target density products, a series of preliminary tests, calculations, and analyses were carried out. Figure 2 shows the photography of GLWC at target densities of 900 and 600 kg/m^3^, and the Al powder percentage added to the corresponding products was 0.04 and 0.2%, respectively.

### 2.3. Test Methods

The bulk densities of lightweight concretes were determined with the Archimedes method, and the water absorption was determined by the Chinese National Standard GB/T 11970-1997 [27]. The compressive strength and flexural strength of GLWC were tested with NYL-300 and DKZ-5000 (Wuxi Jianyi, Wuxi, China) testing machines on 40 mm × 40 mm × 40 mm and 40 mm × 40 mm × 160 mm specimens, respectively. Thermal conductivity measurement was conducted according to Chinese National Standard GB 10294-88 [28].

The microstructural characteristics of geopolymer samples were examined by S-3400N scanning electron microscopy (SEM) (Hitachi, Tokyo, Japan), and the samples were sprayed with gold before testing. Crystalline phase evolutions of powders were analyzed by an X’TRA high-performance powder X-ray diffraction (XRD) (Thermo Electron Corporation, Ecublens, Switzerland) with Cu Kα radiation generated at 40 mA and 40 kV and the step of 0.01° at 5 min^−1^ from 5 to 80°. Thermal analysis (simultaneous thermogravimetric analysis (TG) and differential thermal analysis (DTA) curves) was carried out in order to characterize the nature of the solids on an STA 449C thermal analyzer (Netzsch, Selb, Germany). The sample was heated in an alumina crucible from ambient temperature to 1000 °C in a dry airflow, and the heating rate was kept constant at 10 °C min^−1^. The Fourier Transform Infrared (FTIR) spectra were recorded with a Nicolet 6700 FTIR spectrometer (Thermo Scientific, Raleigh, NC, USA), and samples were compressed with dry high-purity KBr before the spectrum test.

## 3. Results and Discussion

### 3.1. Physical Properties

Density is one of the most important factors that can affect many physical properties of LWC. The density of LWC is mainly determined by the amount of foaming agents. The bulk density of oven-dried samples that were cured for 28 days were measured, and the results are shown in Figure 3. It is obvious from Figure 3 that through controlling Al powder content, GLWCs in a wide range of densities can be produced. The density of GLWC obtained in this study ranges from 500 to 1600 kg/m^3^, which is much lower than traditional OPC concretes. In addition, the bulk density decreased from 1200 to 450 kg/m^3^ with the increase of Al powder content from 0.02 to 0.16%. The bulk density reduced rapidly with an Al powder content increase from 0 to 0.08%. An increase in Al powder content did not have a measurable effect on the bulk density. This might be due to the fact that H_2_ generated by an increase in Al powder cannot be trapped completely within the geopolymer paste. Al powder exhibited a more obvious effect on density than hydrogen peroxide did when it was applied as a foaming agent [29].

The thermal conductivities of GLWC with densities of 600, 900, and 1200 kg/m^3^ cured at 40 °C for 28 days are given in Table 2. The data for commercial aerated concrete are also included for comparison purposes [30]. The thermal conductivities of GLWC synthesized in this study using fly ash as a source material are similar to the results of GLWC based on metakaolin (MK) reported by Aguilar et al. [25] and are better than the results reported by Peng et al. [13]. The thermal conductivity of GLWC is correlated to its density, and the thermal conductivity value can be decreased by changing the cell volume, size, and type. With low thermal conductivity and low bulk density, GLWC is suitable to be used as building blocks or partition walls.

Water absorption can significantly affect the thermal conductivity, permeation resistance, and durability of GLWC. As illustrated in Figure 4, the GLWC samples, especially ones produced with high Al powder content, showed high water absorption. According to [31], water absorption is closely related to the quantity of interconnected pores. With the increase of Al powder content, more H_2_ will be produced. The mitigation of H_2_ bubbles during the hardening process of the geopolymer will induce more interconnected pores, leading to increased water absorption. The water absorption increased from 12.5 to 35% with the increase in Al powder content from 0.02 to 0.12%. In addition, the high surface area resulting from the pores also contributes to the increasing water absorption.

### 3.2. Mechanical Properties

Figure 5 shows the compressive strength of GLWC with various densities after curing for 28 days. The data for commercial aerated concretes are also given for comparison purposes. As shown in Figure 5, the compressive strength drops while the bulk density decreases. Within the range of low density (<600 kg/m^3^), the compressive strength of fly ash GLWC ranges from 1 to 4 MPa, which is similar to the compressive strength of aerated cement. The results could be explained by the high air void content being the decisive factor affecting mechanical strength for the low-density specimens [32]. On the other hand, the GLWCs with a bulk density higher than 600 kg/m^3^ show much higher compressive strength than commercial cement-based LWC. For GLWCs with a bulk density around 900 and 1200 kg/m^3^, the compressive strength is 50 and 30% higher than aerated concrete, respectively. In particular, the average compressive strength of GLWC with densities of 1200 kg/m^3^ is about 13 MPa, which is better than that of commercial aerated concrete. Studies show that LWC manufactured with lightweight aggregate must possess a density of around 1700 kg/m^3^ in order to get strength of about 12 MPa [33]. The compressive strength of cement-based LWC with the density of 800–1000 kg/m^3^ is 1.5–3.0 MPa [34], while the compressive strength of GLWC in that density range is around 6–10 MPa, which is much higher than that of its traditional cement counterparts.

The compressive and flexural strength developments of the selected specimens are presented in Figure 6 and Figure 7. Obviously, the synthesized GLWCs show similar trends in terms of flexural and compressive strength development. GLWCs with different densities develop approximately 80–90% of their corresponding 28 days strength after curing for seven days, and the strength does not have any obvious change after 28 days of curing. This shows that the GLWC reached relatively high compressive and flexural strength in a short time, especially for the samples with densities of 1200 and 900 kg/m^3^. The sample with density of 1200 kg/m^3^ had a compressive strength of about 3 MPa after only one day of curing. This may be attributed to the controllable setting and rapid hardening of geopolymers. This property is beneficial to industrial application in many conditions and makes GLWC more competitive.

### 3.3. Thermogravimetric Property of GLWC

The TG-DTA curves of the selected geopolymers are expressed in Figure 8. It can be seen from Figure 8 that the sample might undergo weight reduction in air with a temperature increase from room temperature to 1000 °C. The results show that about 60% of the mass of all water had been lost at a temperature below 250 °C. The lost water is weakly adsorbed.

The remaining water evaporated at a temperature higher than 250 °C according to the following dehydration reaction [35]:2(SiO_3_^2−^·2M^+^) –OH → (SiO_3_^2−^·2M^+^)_2_·O + H_2_O (↑)

The exothermic peak on the DTA curve above 500 °C may be caused by oxidation of residual carbon that comes from the raw fly ash [36]. Furthermore, the DTA curve of the geopolymer gives a single endothermic peak at 110 °C because of dehydration. No other endothermic peaks for dissolution of hydrates such as Ca(OH)_2_, CaCO_3_, and ettringite were found, which is different in comparison to OPC [37].

### 3.4. Microstructure of GLWC

The microstructural development of GLWC specimens was examined by means of SEM. The SEM images of raw materials as well as geopolymeric lightweight concretes with density of 1200 kg/m^3^ after curing for 28 days are shown in Figure 9. As shown in Figure 9a, the fly ash particles were spherical with an irregular small size. The material structure of LWC is characterized by its solid matrix and large pores. Under low magnification, spherical voids with different diameters introduced by the liberation of H_2_ could easily be observed. Most of the pores are enclosed and un-interconnected due to low content of Al powder in the paste. Figure 9c,d shows that under high magnification, the geopolymers become artificial dense clumps composed of insoluble fly ash particles surrounded by a colloidal inorganic binder. The fly ash particles were further exposed to water and other substances, which can cause the precipitation of alkali or corrode unreacted fly ash. The EDS spectrum in Figure 9e shows high-intensity peaks of silicon (Si), aluminum (Al), and oxygen (O), indicating the main elements of geopolymeric gel. Other elements such as calcium (Ca), sulfur (S), and iron (Fe) are also expressed in the spectrum but with rather low-intensity peaks. Furthermore, the peak distributions of these main elements are also shown in Figure 9e.

### 3.5. Infrared Spectroscopy

The FTIR spectra of the raw fly ash and the obtained geopolymer are summarized in Figure 10. Characteristic bands for Si–O–Si and Si–O–Al asymmetric stretching between 1030 and 1090 cm^−1^, Si–O–Si symmetric stretching around 795 cm^−1^, and Si–O and Al–O in-plane flexural vibration modes near 470 cm^−1^ were identified [38,39]. The broad bands at around 3450 cm^−1^ and 1650–1600 cm^−1^ on the Infrared Radiation (IR) spectrum correspond to O–H stretching and O–H bending, respectively [40]. Compared with the raw fly ash, the Si–O–Si stretching vibration position of the geopolymer at 1095 cm^−1^ shifted to a lower frequency of 1028 cm^−1^ as a consequence of polycondensation with alternating Si–O and Al–O bonds [38], implying a chemical change in the matrix.

### 3.6. Mineralogical Analyses

The XRD analysis results of raw fly ash and final GLWC are shown in Figure 11. From this picture, it can be seen that a broad hump at around 20–40° exists on the XRD patterns of raw fly ash indicating there is a large amount of amorphous phase. Besides this, strong crystalline peaks of quartz and mullite were also easily identified. The XRD patterns of the geopolymer do not show any remarkable changes from that of fly ash. After the GLWC samples were cured for 90 days, no new crystalline peaks could be found in the XRD pattern, and the peaks of quartz and mullite were still intense. The results indicated that only the amorphous phase of the fly ash participated in the reaction of geopolymerization. With that said, it was not possible to detect Al powder considering the low amount added and the reaction with alkali.

## 4. Summary and Conclusions

GLWC was synthesized using fly ash as a solid precursor via geopolymer technology, and Al powder was applied as an in situ foaming agent. The main findings can be summarized as follows:(1)GLWC provides a better performance than traditional cement-based LWC, especially in terms of mechanical strength. Geopolymers can provide high compressive strengths when the density is reduced, and their mechanical properties are acceptable.(2)Apart from possessing relatively higher thermal conductivity than OPC-based LWC, the thermal conductivity and strength of GLWC synthesized in this study also reached a better balance.(3)The optimal GLWC produced in this study is the one with a density of 900 kg/m^3^; its 28 days compressive strength is 10 MPa, and its thermal conductivity is 0.52 W/mK, both of which are higher than those of its OPC counterpart of similar density.(4)In contrast to OPC, the main mineralogical phases of GLWC contain various amorphous and semicrystalline phases with a decent microstructure.

## Figures and Tables

**Figure 1 materials-13-04059-f001:**
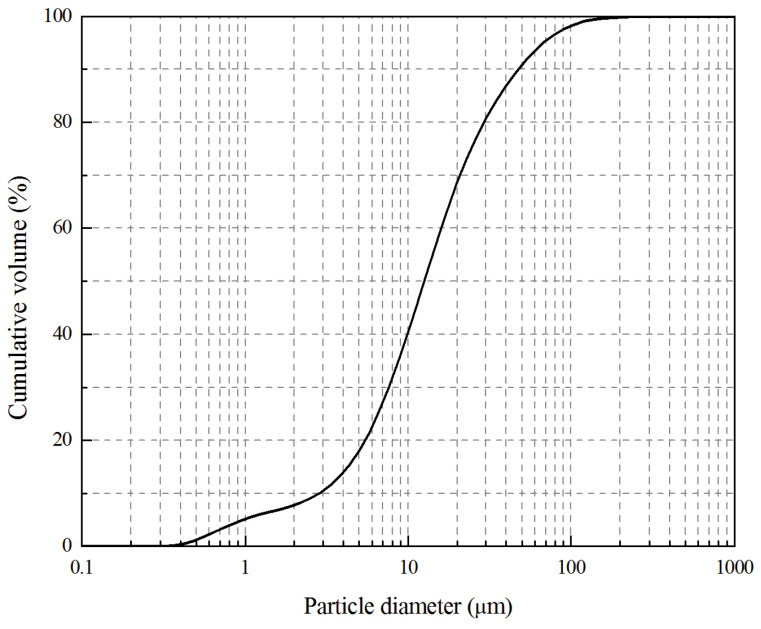
Particle size distribution of coal fly ash.

**Figure 2 materials-13-04059-f002:**
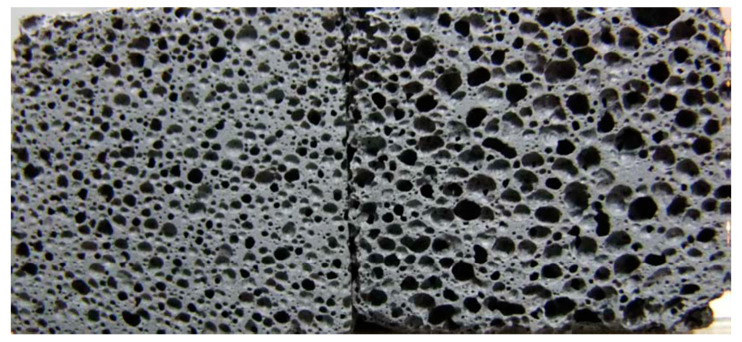
Photography of polished sections of GLWC of 900 kg/m^3^ (left) and 600 kg/m^3^ (right). The length and width of GLWC are 4 cm.

**Figure 3 materials-13-04059-f003:**
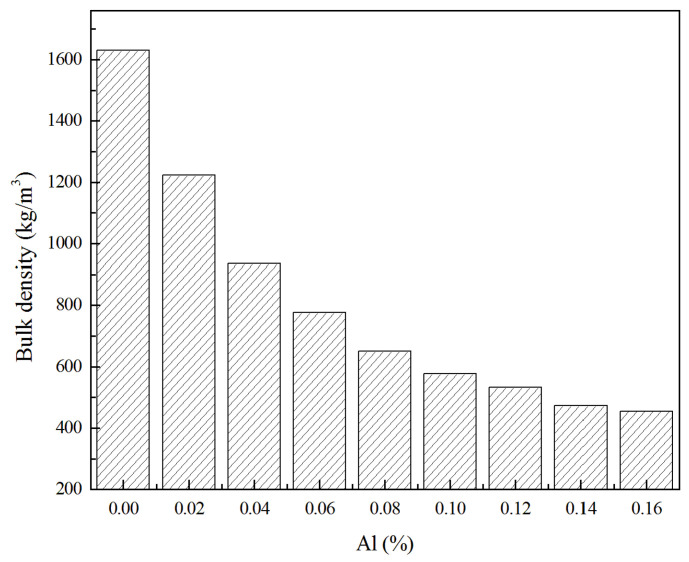
Bulk density as a function of Al powder content.

**Figure 4 materials-13-04059-f004:**
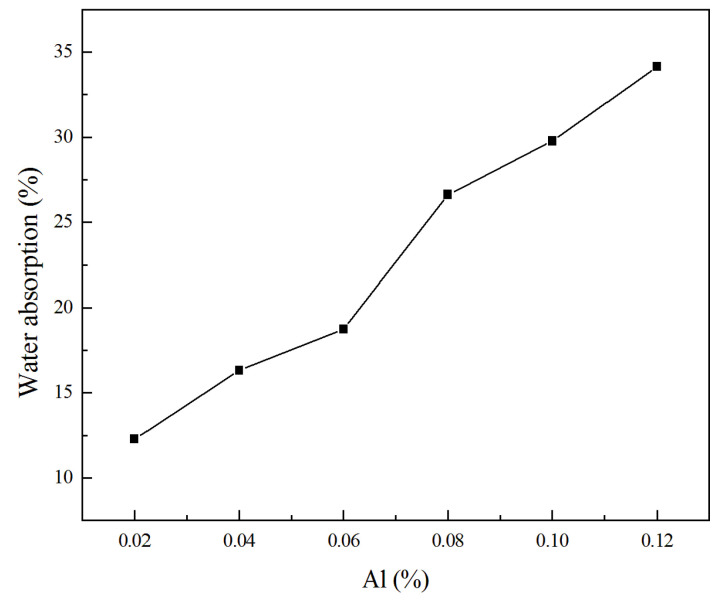
Water absorption of GLWC vs. Al powder content.

**Figure 5 materials-13-04059-f005:**
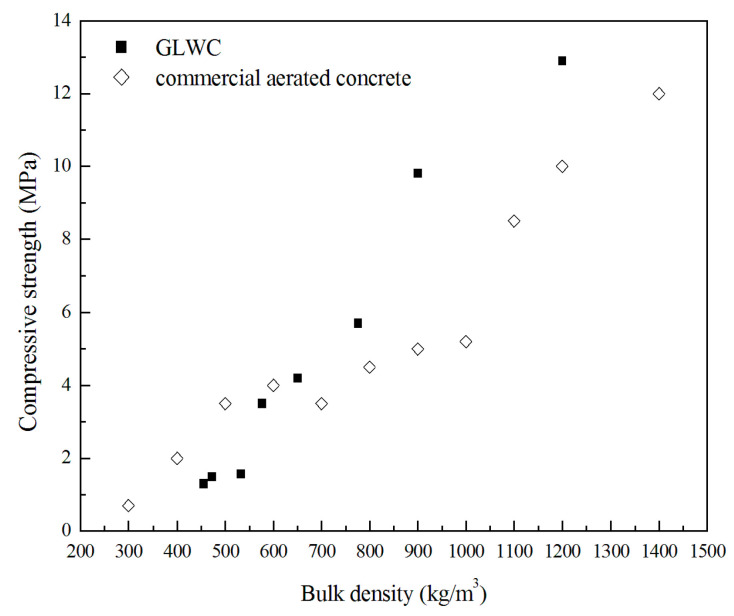
Compressive strength vs. bulk densities of GLWC after curing for 28 days.

**Figure 6 materials-13-04059-f006:**
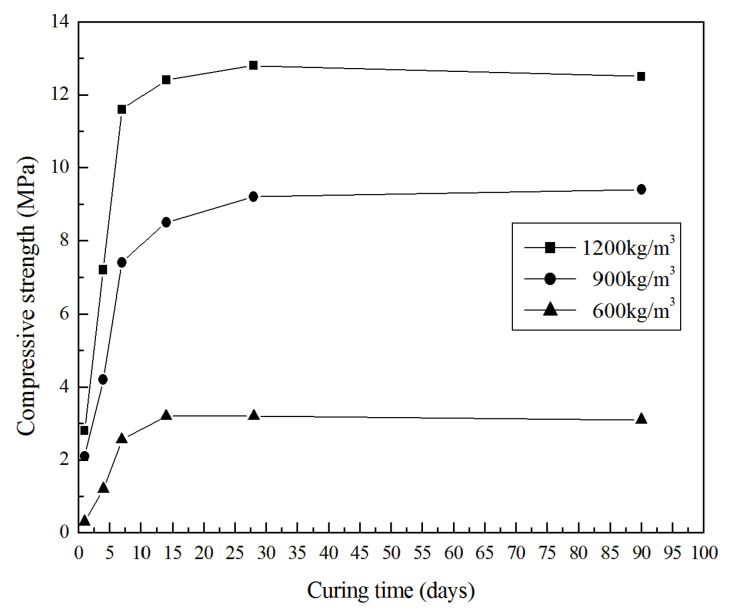
Compressive strength of GLWC vs. curing time.

**Figure 7 materials-13-04059-f007:**
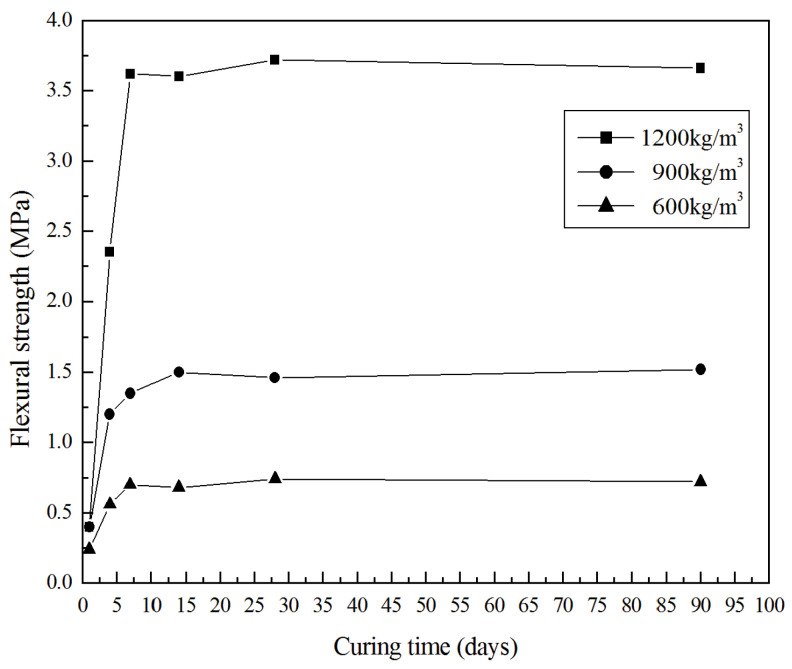
Flexural strength of GLWC vs. curing time.

**Figure 8 materials-13-04059-f008:**
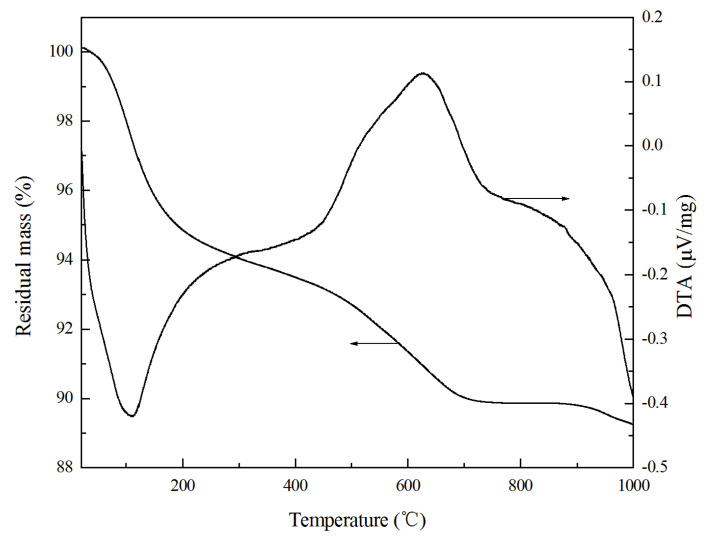
TG-DTA curves of selected GLWC curing for 28 days.

**Figure 9 materials-13-04059-f009:**
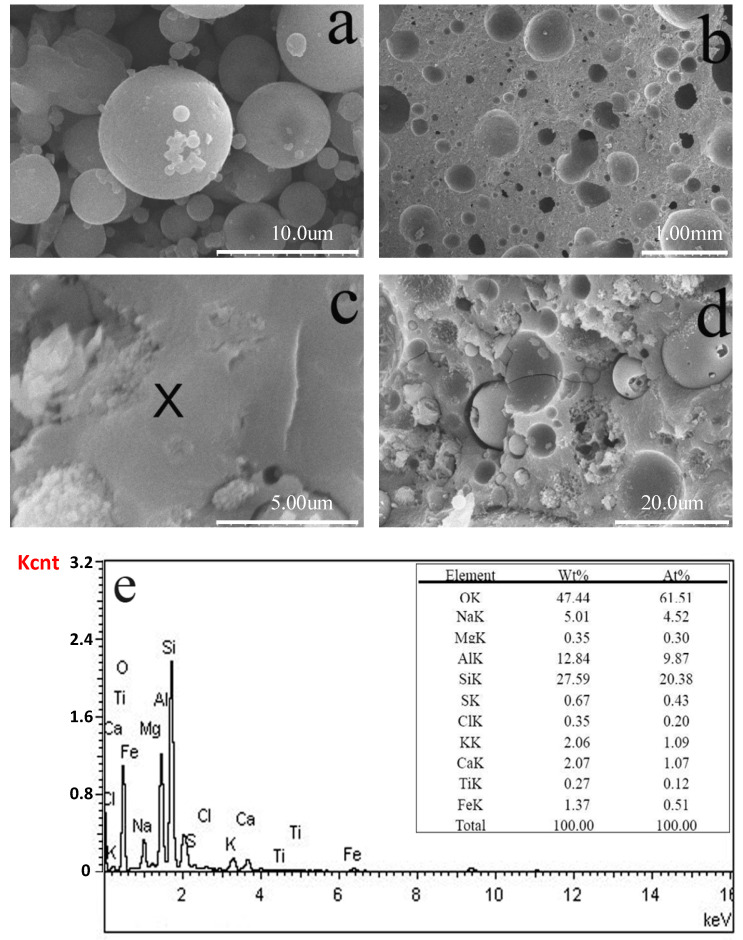
SEM–EDS (Energy Dispersive Spectrometer) characterization of fly ash and GLWC after curing for 28 days. (**a**) SEM microphotography of raw fly ash; (**b**,**c**,**d**) SEM microphotography of GLWC; (**e**) chemical characterization by EDS of the sample area marked as X on the photograph.

**Figure 10 materials-13-04059-f010:**
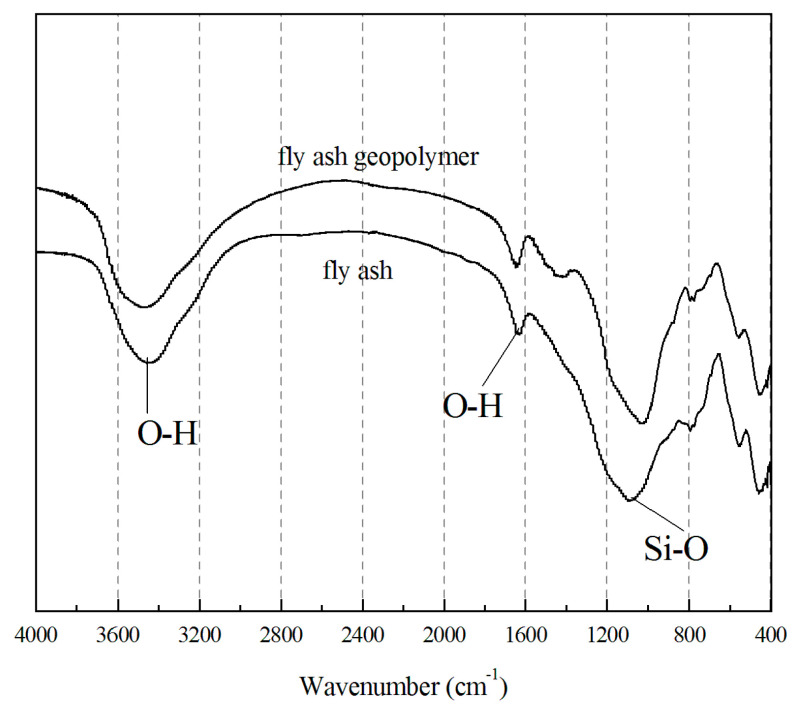
FTIR spectra of raw fly ash and geopolymeric lightweight concretes.

**Figure 11 materials-13-04059-f011:**
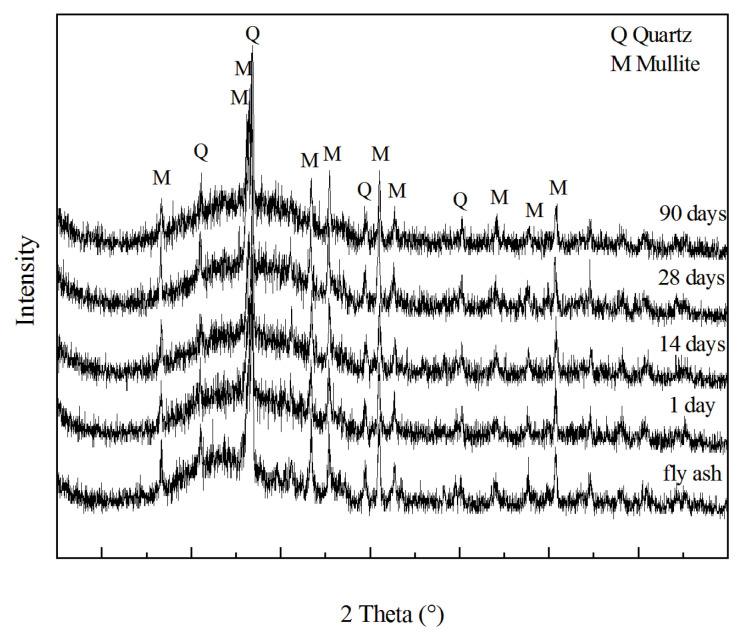
X-ray diffraction (XRD) patterns of fly ash and GLWC samples after curing for 1, 14, 28, and 90 days.

**Table 1 materials-13-04059-t001:** Oxide composition of the fly ash.

Component	Composition (mass %)
Al_2_O_3_	28.62
SiO_2_	53.18
CaO	3.83
Fe_2_O_3_	5.64
K_2_O	1.60
MgO	0.84
Na_2_O	0.65
P_2_O_5_	0.18
TiO_2_	1.37
LOI ^a^	3.41

^a^ Loss on ignition (LOI) at 960 °C.

**Table 2 materials-13-04059-t002:** Thermal conductivity values of GLWC and commercial cement-based lightweight concrete (LWC).

Specimen Density (kg/m^3^)	Thermal Conductivity (W/mK)	
	GLWC	Commercial LWC [25]
600	0.22	0.115
900	0.52	0.175
1200	0.70	0.270

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
