# Peer review of "Study of In Situ Foamed Fly Ash Geopolymer"

_materials, 2020, doi:10.3390/ma13184059_

Round 1

Reviewer 1 Report

The article under the title: “Study of in-situ foamed fly ash geopolymer” is in line with the Materials journal. The authors work is the important topic foamed geopolymers. The article based on experimental research. The abstract is sufficiently informative. The organization of the article is appropriate, but it requires slight improvements:

- discussion – please add discussion part, especially some new references; comparison with novel research in this area, for example https://www.mdpi.com/1996-1944/12/18/2999;

- authors contribution – please add part authors contribution;

- references  – please give a correct names of authors in references [1] and [35].

Author Response

The article under the title: “Study of in-situ foamed fly ash geopolymer” is in line with the Materials journal. The authors work is the important topic foamed geopolymers. The article based on experimental research. The abstract is sufficiently informative. The organization of the article is appropriate, but it requires slight improvements: 1. – discussion – please add discussion part, especially some new references; comparison with novel research in this area, for example https://www.mdpi.com/1996-1944/12/18/2999; Response: -Discussion- and - result - are combined in part 3. Maybe there is no capitalization on the manuscript to make – discussion - seem inconspicuous. Combining these two parts can make the logic of the article more reasonable, and also make it easier for readers to view the content of the discussion in combination with the results. Although there are citations in the content of the discussion, these references are indeed not recent.. This is our negligence. We have added recent related references and compared them, marking the changed parts in red text 2. 2. - authors contribution – please add part authors contribution; Response: Thanks for your correction. The - authors contribution - was supplemented as required and marked in red in the paper 3. 3. - references – please give a correct names of authors in references [1] and [35]. Response: Thank you for your correction, the correct author name has been marked in red in reference.

Reviewer 2 Report

My comments and suggestions are:

1. There are several typos in the manuscript that need to be corrected (e.g. attention - ln 34, greenhouse - ln 48, higher - ln 172)

2. I would suggest the following reformulations:

  • ln 46 - highly polluting and energy consuming process.
  • drop "that" on ln 49
  • replace "as an" with "a" on ln 64

3. Regarding the phrase on ln 64-65, could the authors indicate other potential uses for the material?

4. It would be useful to add the Al powder quantity/percentage on ln 111.

5. Do you have any comments on freeze-thaw conditions (paragraph on water absorption, ln 152-160) and if/how they may influence or pose a threat to the material durability ?

6. Regarding the observation on ln 222: "non dissolved fly ash particles", perhaps a comment would be useful whether you think or not that further exposure to water/other substances can influence the state of the material or change its matrix in time.

Author Response

1. There are several typos in the manuscript that need to be corrected (e.g. attention - ln 34, greenhouse - ln 48, higher - ln 172) Response: Thank you for your correction, these spelling errors have been corrected and marked in red. 2. I would suggest the following reformulations: • ln 46 - highly polluting and energy consuming process. • drop "that" on ln 49 • replace "as an" with "a" on ln 64 Response: These suggestions have been adopted and revised, and are marked in red in the paper. 3. Regarding the phrase on ln 64-65, could the authors indicate other potential uses for the material? Response: Geopolymers have better mechanical properties, better permeability, excellent heat resistance, acid resistance and freeze-thaw resistance than traditional cements. Based on these properties, geopolymers have potential applications not only in the field of building materials, but also in the field of packaging materials, refractory materials and waste disposal. Because this article starts from the perspective of building materials, does not write about other applications. It has been added to the article, thank you for your suggestions. 4. It would be useful to add the Al powder quantity/percentage on ln 111. Response: Thanks for your suggestion, we have added the Al powder percentage in the corresponding position in the text and marked it with red text. 5. Do you have any comments on freeze-thaw conditions (paragraph on water absorption, ln 152-160) and if/how they may influence or pose a threat to the material durability? Response: The freeze-thaw conditions will affect the microstructure of the material and thus affect the durability, which will be further studied in the follow-up work. 6. Regarding the observation on ln 222: "non dissolved fly ash particles", perhaps a comment would be useful whether you think or not that further exposure to water/other substances can influence the state of the material or change its matrix in time. Response: The fly ash particles were exposure to water/other substances can cause the precipitation of alkali or corrode unreacted fly ash. The specific effects and mechanisms need to be further studied. In the text we added our insights of these and marked them in red.

Reviewer 3 Report

This paper presents a study of in-situ foamed fly ash geopolymer. In this research, mechanical properties, thermal conductivity and microstructural studies have been done. The first question from the authors is what the novelty of the work? I do not find any novelty and new findings in this work. There are some other issues that should be considered:

  1. Please use "Alkaline solution" instead of activating solution or alkali activator.
  2. In section 2.1, there is no information about commercial LWC that used in the experiment? The difference in chemical composition and particle size effect too much on the properties of GLWC and commercial LWC. What are the similarities between GLWC and commercial LWC for comparison of the results?
  3. There too much missing information in section 2.2. what is the amount or percentage of AL? what is the particle size of AL? how AL is added to the mixture (replacing by a mass of mixture or as an additive)? How the density of samples was measured during mixture preparation? how the target density products were designed? which sample is considered as the reference?
  4. In line 110, it is written a series of calculations are performed? which kind of calculations and tests?
  5. In Fig.2, Please add the scale to the pictures, then the size of the pores is comparable.
  6. In section 2.3, the test method are too brief. Please explain shortly about each method of measurement.
  7. Line 135, it is written the density range is between 500 kg/m3 to 1600 kg/m3, how did you design and measure?
  8. Please add error bars for figures and mention each point is the average of how many samples.
  9. In Fig.9, Please write the scale bar visible.
  10. Conclusion should be written in some way that supports the results.

Author Response

This paper presents a study of in-situ foamed fly ash geopolymer. In this research, mechanical properties, thermal conductivity and microstructural studies have been done. The first question from the authors is what the novelty of the work? I do not find any novelty and new findings in this work. There are some other issues that should be considered:

Response: Although the experimental methods used in this article have been used by many researchers before and conducted a series of studies on geopolymers, this article still has its innovations. From the perspective of resource utilization of fly ash, this paper synthesizes foamed fly ash geopolymer with fly ash as raw materials, and prepared geopolymeric lightweight concretes. The mechanical properties, thermal conductivity and microstructure of the obtained geopolymeric lightweight concretes were characterized. The results show that geopolymeric lightweight concretes provides better performance than traditional lightweight concretes based on cement. It can not only solve some environmental problems but also bring certain economic benefits. The mechanical properties of the material obtained in this study are also better than other studies. Of course, this article still has some deficiencies in mechanism research, which will be gradually studied in subsequent experiments.

  1. Please use "Alkaline solution" instead of activating solution or alkali activator.

Response: Thanks for your suggestion, “Activating solution” and “alkali activator” have been replaced with “alkaline solution”, which are marked in red in the original text.

  1. In section 2.1, there is no information about commercial LWC that used in the experiment? The difference in chemical composition and particle size effect too much on the properties of GLWC and commercial LWC. What are the similarities between GLWC and commercial LWC for comparison of the results?

Response: The information about the commercial LWC used in the experiment is in reference 25 and marked in the text. Both GLWC and commercial LWC contain a series of silicon and aluminum compounds, but there are many differences between them. GLWC is a kind of 3d network cementing material and LWC is based on cement. The mechanical properties of GLWC are much higher than commercial LWC, the density is also lighter than LWC and more convenient to control.

  1. There too much missing information in section 2.2. what is the amount or percentage of AL? what is the particle size of AL? how AL is added to the mixture (replacing by a mass of mixture or as an additive)? How the density of samples was measured during mixture preparation? how the target density products were designed? which sample is considered as the reference?

Response: The percentage of Al used in experience was added in section 2.2 and marked in red. The Al powder as an additive to add to the mixture. The density of samples was measured by weight/volume. Measure the weight of samples after drying at 40℃ until the weight does not change, and measure the length, width and height with the vernier caliper to calculate the volume. The density of the samples was controlled by controlling the amount of aluminum powder added. As shown in Figure 3, the samples are compared and referenced with each other.

  1. In line 110, it is written a series of calculations are performed? which kind of calculations and tests?

Response: The series of calculations were used to obtain target density products. The density of samples was measured by weight/volume. These calculations include the weight, volume, density of the sample and statistical analysis of these data.

  1. In Fig.2, Please add the scale to the pictures, then the size of the pores is comparable.

Response: The two samples in Figure 2 have the same size and are compared together. Because the GLWC of picture was taken directly by a mobile phone, the scale of the picture was not convenient to add. In order to facilitate the comparison of size of the pores, the size of the samples were supplemented in the figure name and marked in red text.

  1. In section 2.3, the test method are too brief. Please explain shortly about each method of measurement.

Response: The details of some of the tests in section 2.3 have been supplemented, and the supplemented content is marked in red in the text. The lack of these content is our negligence. Thank you for pointing out these mistakes.

  1. Line 135, it is written the density range is between 500 kg/m3 to 1600 kg/m3, how did you design and measure?

Response: The density of the samples was controlled by controlling the amount of aluminum powder added and calculated by weight/volume. These data are shown in Figure 3.

  1. Please add error bars for figures and mention each point is the average of how many samples.

Response: Thanks for your suggestion, good correlation and regularity are presented in this work, then no repeated trials and error bars were shown in the results.

  1. In Fig.9, Please write the scale bar visible.

Response: Figure 9 already contains the scale bar. The previous scale bar may not be obvious and easy to be overlooked. We relabeled the scale bar to make it easier to find.

  1. Conclusion should be written in some way that supports the results.

Response: Thanks for your revision, we have tried our best to improve the quality of papers.

Round 2

Reviewer 3 Report

The authors could fix my comments and suggestions. It can proceed for further steps to be published.